# Study on the Microstructure and Properties of FeCoNiCrAl High-Entropy Alloy Coating Prepared by Laser Cladding-Remelting

Tianyi Lv, Wenkai Zou, Jiaqi He, Xiang Ju and Chuanbo Zheng *

School of Metallurgy and Materials Engineering, Jiangsu University of Science and Technology, Zhenjiang 212100, China; ltyyyasmy@gmail.com (T.L.); z984731733@gmail.com (W.Z.); 17348673267@163.com (J.H.); 18232560007@163.com (X.J.)

* Correspondence: 15952802516@139.com

**Abstract:** Laser remelting technology effectively repairs defects such as pores and cracks in the coating. To investigate the impact of laser remelting on high-entropy alloy coatings, this study used Q235 steel as the substrate and employed laser cladding technology to prepare FeCoNiCrAl high-entropy alloy coatings, followed by laser remelting treatment. The phase composition and microstructure of the coatings were extensively characterized using equipment such as optical microscopy, X-ray diffraction (XRD), and scanning electron microscopy (SEM). Additionally, the wear resistance and corrosion resistance of the coatings were tested using a multifunctional material surface performance tester, an electrochemical workstation, and SVET (Scanning Vibrating Electrode Technique). The results indicate that following laser remelting treatment, the atomic proportion of Fe elements on the coating surface decreased from 33.21% to 26.03%, while the atomic proportion of Al elements increased from 12.56% to 20.31%. The phase composition of the coating underwent a marked transformation, shifting from a structure composed of FCC, A2, and B2 phases to a singular BCC structure characterized by the presence of A2 and B2 phases. Concurrently, the grain morphology on the coating surface transitioned from elongated plate-like grains to equiaxed grains. Laser remelting enhanced the wear resistance of the coating. Laser remelting had no significant impact on the corrosion resistance of the non-cracked regions of the coating.

**Keywords:** high-entropy alloys coatings; laser cladding; laser remelting; corrosion; microstructure

## 1. Introduction

As the industrial sector advances, the demand for metal materials in industrial settings has become increasingly rigorous. Merely altering the composition and structure of the material is insufficient to fully enhance numerous properties of the material. By preparing a protective coating on the surface of the matrix material, the properties of the matrix material are maintained and the abrasion and corrosion resistance of the material in the service environment is improved [1]. The combination of high entropy, slow diffusion, lattice distortion, and cocktail effects such as excellent corrosion resistance [2], high microhardness [3], good abrasion resistance [4], and excellent high-temperature antioxidant properties [5] makes high-entropy alloys a promising option for protective coating additive manufacturing. Different techniques can be employed to create high-entropy alloy coatings, such as magnetron sputtering [6], cold spraying [7], plasma spraying [8], high-speed oxygen fuel spraying [9], electrochemical deposition [10], and laser cladding [11]. Laser cladding has a strong bond with the substrate, making it a popular choice for the fabrication and exploration of high-entropy alloy coatings [11].

Rapid heating and cooling during laser cladding causes thermal stress to the coating material. Various components of the material may enlarge and shrink at varying speeds, resulting in fissures appearing on the surface [12]. Undoubtedly, repreparing the coating

increases costs, so there is an urgent need to find a cost-effective solution. In order to repair surface defects and enhance material properties, laser modification technology is widely applied in the field of materials [13]. Laser remelting is a commonly used technique in laser modification. By combining laser rapid solidification and chemical (Sr) modification, Ghosh et al. [14] synthesized a fully eutectic Al-Si microstructure with heavily twinned silicon nanofibers, which has high hardness up to 2.9 GPa and high compressive flow strength (~840 MPa) with a plastic flow that stabilizes to approximately 26% plastic strain. Wu et al. [15] combined the advantages of laser remelting and micro-arc oxidation to prepare a graded structure wear-resistant coating on the surface of titanium alloy, enhancing the wear resistance of the titanium alloy surface. Wang et al. [16] used laser cladding and laser remelting to fabricate iron-based amorphous coating on H13 steel. The results showed that the cracks and stomata defects on the coating decreased significantly. Jin et al. [17] prepared iron-based amorphous and nanocrystalline coatings by plasma spraying and performed laser remelting. Chen et al. [18] improved the density and surface quality of $FeCrCoNiAl_{0.6}$ high-entropy alloy coatings; in order to do this, the coating was prepared and laser remelted using high velocity oxygen fuel (HVOF). The results showed that the friction coefficient of the remelting coating was significantly lower than that of the HVOF coating and no sputtering stratification or cracks were observed in the remelted coating. Some studies indicate that laser remelting technology also enhances the corrosion resistance of the material surface. Al-Sayed et al. [19] conducted laser surface treatment on Ti-6Al-4V alloy. Laser treatment produced a transformed layer with acicular martensite $\alpha'$-Ti as a result of the recrystallization of the fully lamellar $\alpha + \beta$ microstructure of the substrate. The results indicated that ultra-short pulses (in nanosecond scale) from a Nd:YAG laser effectively enhanced the corrosion resistance of this alloy in acidic media (Ringer's solution). Liu et al. [20] conducted laser remelting treatment on the surface of TA15 alloy. The results indicate that as energy density increases, the corrosion rate decreases first and then increases. With an energy density of 8.75 J·cm$^{-2}$, the surface corrosion rate was 20.43 times slower than that of the untreated sample.

Currently, there is a lack of research into laser remelting to improve the coating of high-entropy alloy. Further investigation is necessary to determine the alterations in the element distribution, microstructure, and properties of the coating following remelting. AlCoCrFeNi HEA systems have attracted increasing attention due to their light weight and low cost [21]. Coaxial powder feeding laser cladding was used to create a high-entropy alloy coating of FeCoNiCrAl on Q235 steel in this study. In this study, the Scanning Vibrating Electrode Technique (SVET) was employed in addition to the kinetic potential and electrochemical impedance techniques typically utilized in prior research to gauge and analyze the corrosion potential produced by the microregion of the remelted coating. The influence of remelting on the corrosion resistance of the coating was further explored due to SVET's capability to display local cathode and anode reactions [22].

## 2. Experimental Procedures

### 2.1. Materials

Laser cladding was employed to create a coating of FeCoNiCrAl high-entropy alloy. Using a Q235 plate of 130 mm × 130 mm × 10 mm as a substrate, the cladding material was FeCoNiCrAl High-Entropy Alloy Powder with 53–105 μm diameter produced by vacuum atomization of cladding. (See Figure 1).

### 2.2. Laser Cladding and Laser Remelting Process

Before the thermal spraying process, the oxide film on the substrate surface was polished using a grinding wheel, and any grease was removed by cleaning with anhydrous ethanol. The substrate was preheated to 200 °C, with a laser power of 1200 W, a scanning speed of 300 mm/min, a spot diameter of 4 mm, and the use of coaxial powder feeding at a rate of 20 g/min. The entire process was carried out under an argon gas shield with a flow rate of 2 L/min. The laser remelting was performed at a power of 800 W, a scanning

speed of 500 mm/min, and a spot diameter of 4 mm. To reduce surface roughness and ensure uniformity, the scanning direction was perpendicular to the direction of the thermal spraying [23], as shown in Figure 2:

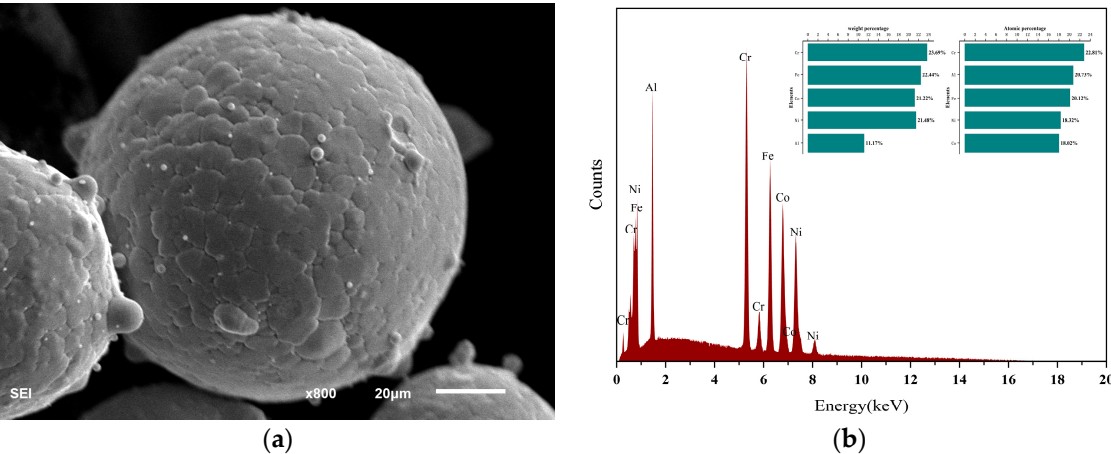

**Figure 1.** FeCoNiCrAl high-entropy alloy powder: (**a**) micromorphology; (**b**) element content.

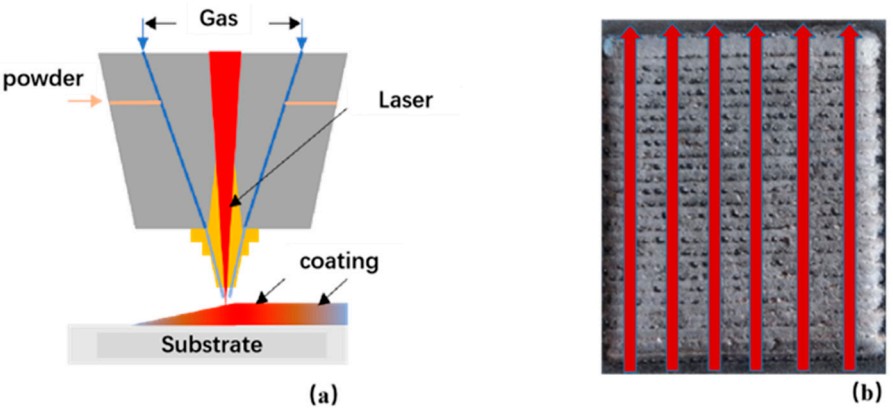

**Figure 2.** Schematic representation of laser cladding and remelting: (**a**) laser cladding with coaxial powder feeding; (**b**) laser remelting direction.

### 2.3. Characterization

Following the process of cutting the laser cladding and remelting the coating to a thickness of 10 mm × 10 mm × 10 mm, sanding and polishing the coating's cross-section using wire cutters, and corroding it with king water (hydrochloric acid: nitric acid = 3:1), the microstructure and phase composition of the coated samples were subsequently examined using optical microscopy (Axio Scope A1), scanning electron microscopy (JSM-6510LA), and an Ultima IV X-ray diffractometer (XRD). The X-ray diffraction analysis focused on the Cu target, which had a scanning rate of 0.02°/s and a scanning angle from 20° to 80°.

An HVS-1000A Micro Vickers scale hardness meter was used to measure the hardness of the coating cross-section, which was then measured at a distance of 0.1 mm from the substrate to the top of the coating. The coating had a payload of 200 g and a retention time of 15 s. The friction performance of the coating was assessed using the MFT-4000 multifunctional material surface performance tester.

ZenniumE4 electrochemical workstations were used to conduct electrochemical corrosion experiments on the samples, with the saturated glycerine electrode as the reference electrode and the platinum electrode as the three-electrode measurement system. A Princeton VersaSCAN Microscan Electrochemical Workstation (AMETEK, Inc., Berwyn, PA, USA) was utilized for microelectrochemical testing, with potentiometric scanning ranging from −1 to 1.5 V, frequency ranging from 0.01 to $10^5$ HZ, and scanning speed of 0.001 V/s. An

investigation was conducted on the microcorrosion of coatings in SVET technology. The frequency selection was set at 5 KHz, the scanning area spanned 1 mm × 1 mm, and the step length measured 40 μm.

## 3. Results

### 3.1. Pahse

The XRD spectra of FeCoNiCrAl HEA powders and coatings are shown in Figure 3. The powders exhibit a single BCC structure, including an Al-Ni-rich BCC(B2) phase, and a Fe-Cr rich disordered BCC(A2) phase. The HEA coating prepared by laser melting contains FCC, A2, and B2 phases. After the coatings were treated by laser remelting, the diffraction peaks of the phase representing FCC disappeared, and only the peaks of the A2 and B2 phases were detected in the XRD tests. Based on the Bragg equation, the lattice parameters of the FCC, A2, and B2 phases were calculated as shown in Table 1.

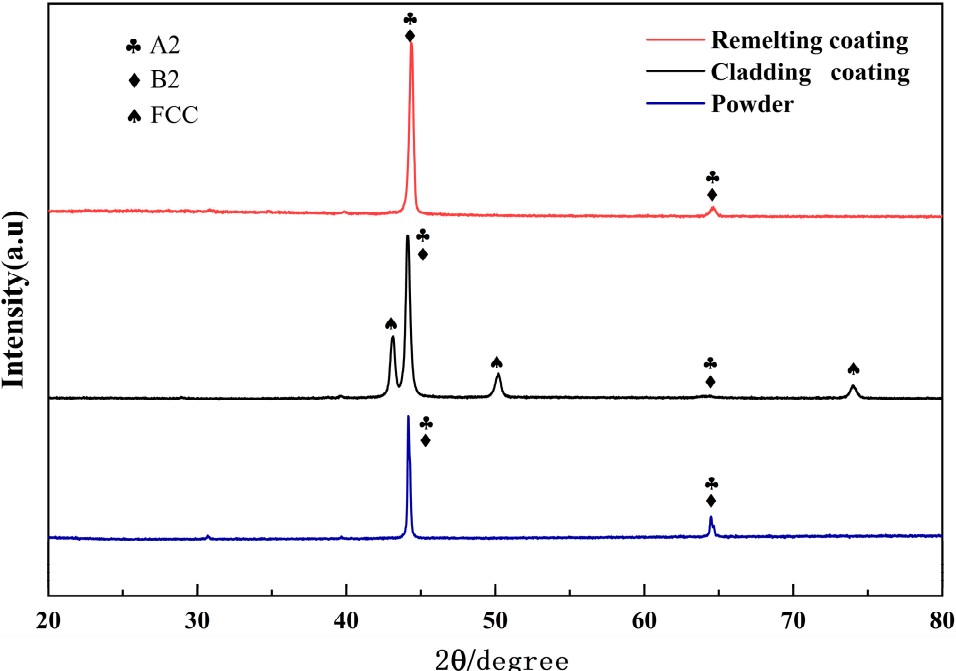

**Figure 3.** XRD diffraction pattern.

**Table 1.** Lattice parameter (Å).

| Sample | FCC | A2 | B2 |
|---|---|---|---|
| Cladding coating | 3.581 | 2.872 | 2.871 |
| Remelting coating | | 2.875 | 2.873 |

### 3.2. Microstructure Characterization

Figure 4 shows the cross-sectional morphology of laser cladding under an optical microscope, and there are apparent bonding lines between the coating and substrate. The bottom and middle parts of the coating are mainly dominated by dendritic crystal distribution, containing a small number of columnar crystals. As the distance from the substrate increases, columnar crystals and equiaxed crystals begin to appear in the upper and middle parts of the coating. Figure 5 shows the EDS scan of the elemental content of the coating cross-section. From the line scan results, it can be seen that during the laser cladding process, due to the dilution effect, the Fe in the coating diffuses into the coating, and with the increase in the distance from the substrate, the Fe elemental content decreases gradually, and the content of the Al element in the coating increases, thus promoting the transformation from dendritic crystals to equiaxed grains. (See Table 2).

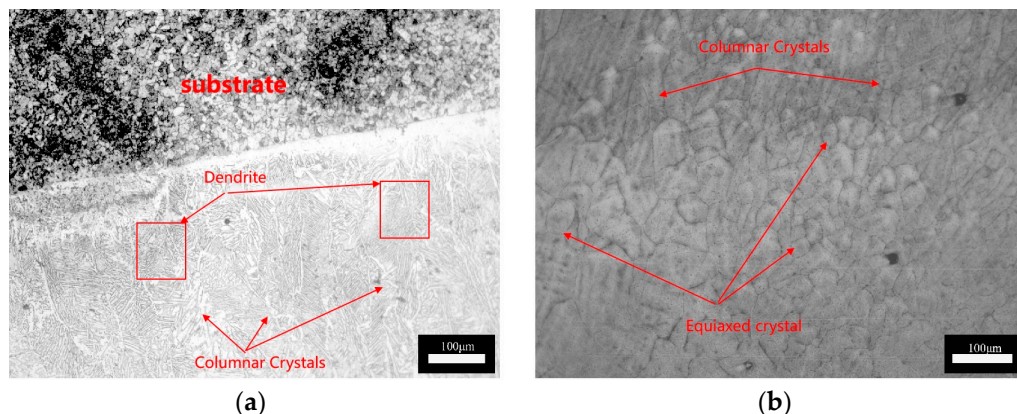

(**a**)　　　　　　　　　　　　　　　　(**b**)

**Figure 4.** Cross-sectional microstructure of the coating: (**a**) midsection of the coating; (**b**) top section of the coating.

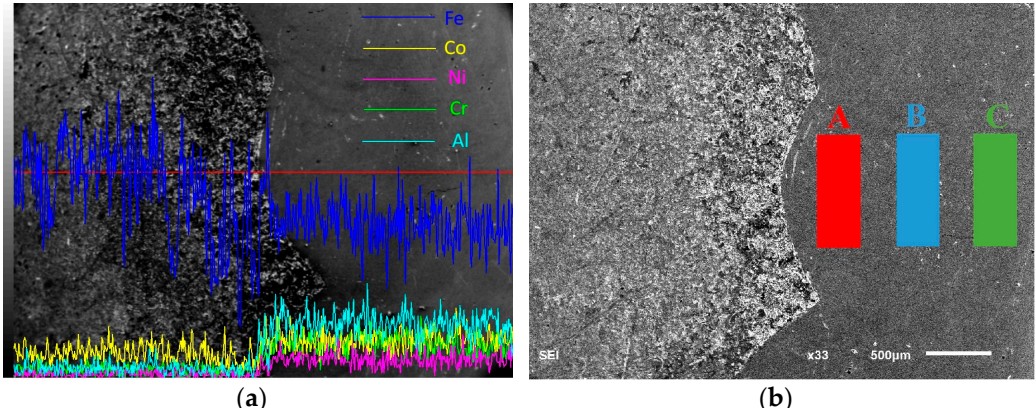

(**a**)　　　　　　　　　　　　　　　　(**b**)

**Figure 5.** Elemental distribution in the cross-section of the coating: (**a**) linear scanning (**b**) selected area scanning.

**Table 2.** EDS results of FeCoNiCrAl high-entropy alloy coatings (At%).

|   | Fe | Co | Ni | Cr | Al |
|---|------|------|------|------|------|
| A | 39.77 | 14.96 | 13.55 | 14.69 | 17.03 |
| B | 32.50 | 18.13 | 12.74 | 16.31 | 20.31 |
| C | 31.65 | 17.46 | 17.39 | 14.65 | 18.85 |

Figure 6a shows the micromorphology of the surface of the FeCoNiCrAl high-entropy alloy coating prepared by laser melting cladding, which is mainly distributed with elongated plate-like grains and some fine grains. After the coating was laser remelted as shown in Figure 6b, the grain structure of the coating surface was transformed into equiaxed grains. To determine whether there is any elemental segregation on the coating surface, the coating before and after remelting was analyzed by surface scanning, and the results are shown in Figures 7 and 8. It can be seen that there is no obvious elemental segregation in the coating before and after remelting, and the five elements of Fe, Co, Ni Cr, and Al are uniformly distributed in the grain boundaries. However, the proportion of each element in the coating changed significantly after remelting. After remelting, the atomic ratio of Fe decreased from 33.21% to 26.03% and the atomic ratio of Al increased from 12.56% to 20.31%, which may be the reason for the transformation of the grain structure of the coating surface.

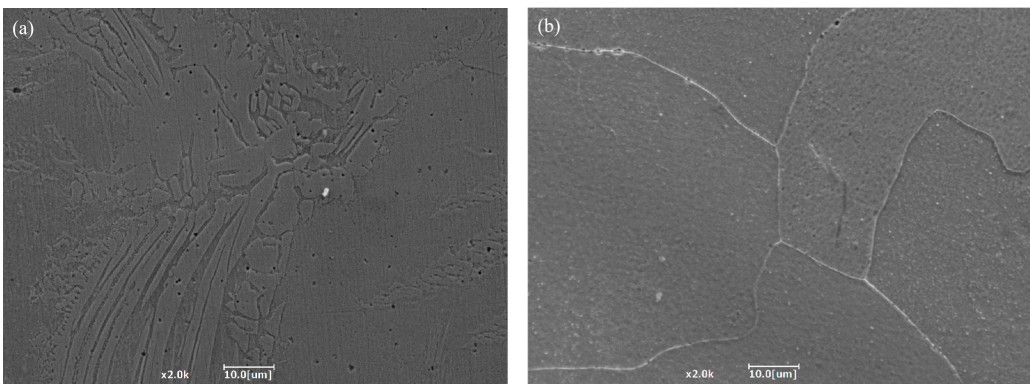

**Figure 6.** SEM images of the coating surfaces: (**a**) cladding coating; (**b**) remelting coating.

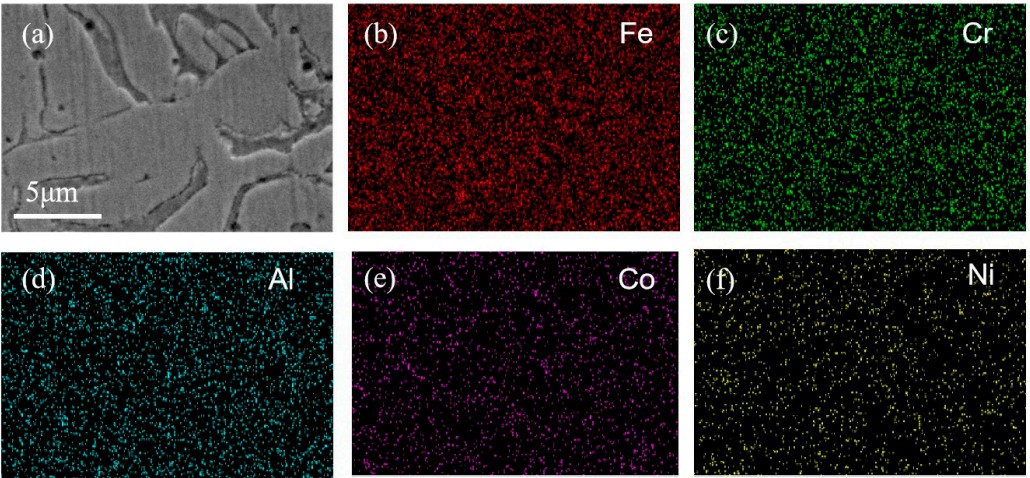

**Figure 7.** EDS mapping of FeCoNiCrAl high-entropy alloy coatings: (**a**) cladding coating; (**b**) Fe; (**c**) Cr; (**d**) Al; (**e**) Co; (**f**) Ni.

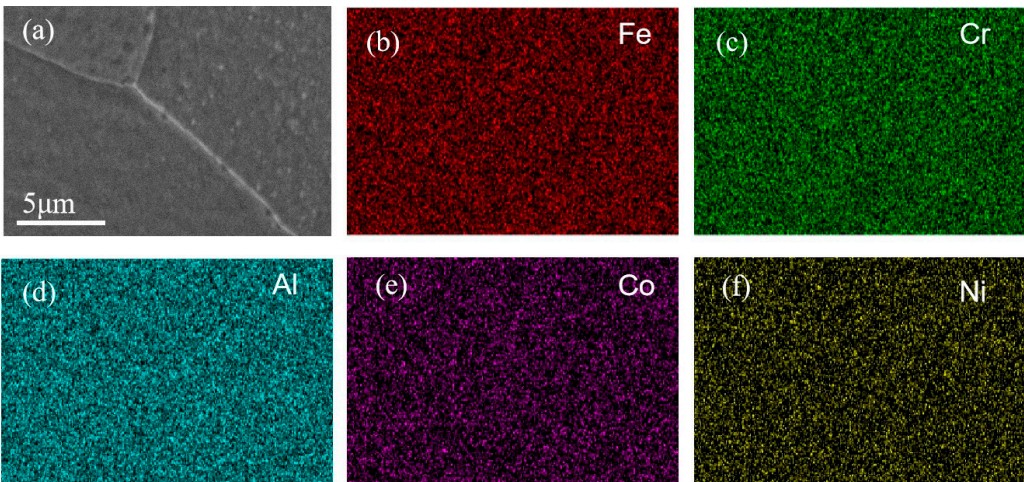

**Figure 8.** EDS mapping of FeCoNiCrAl high-entropy alloy coatings: (**a**) remelting coating; (**b**) Fe; (**c**) Cr; (**d**) Al; (**e**) Co; (**f**) Ni.

### 3.3. Microhardness and Wear Resistance of Coatings

Figure 9 shows the microhardness distribution of the laser cladding coating and laser remelting coating. The hardness test is conducted at 12 point intervals, with each interval measuring 0.1 mm from the junction of the coating and substrate, perpendicular to the

direction of the cross-section to the coating's surface. The diagram indicates that the coating and substrates have a hardness of approximately 250 to 300 $Hv_{0.2}$ at the junction and 500 to 600 $Hv_{0.2}$ at a distance of 0.4 mm to 0.6 mm from the coating's surface. The remelting area experiences a gradual increase in microhardness, specifically within the range of 0 to 0.4 mm from the coating's surface. The coating exhibited a microhardness of 613 $Hv_{0.2}$ and 754 $Hv_{0.2}$, correspondingly, until it underwent fusion and subsequent remelting. The laser-remelted coating exhibited a 23% increase in hardness in comparison to the non-remelted coating.

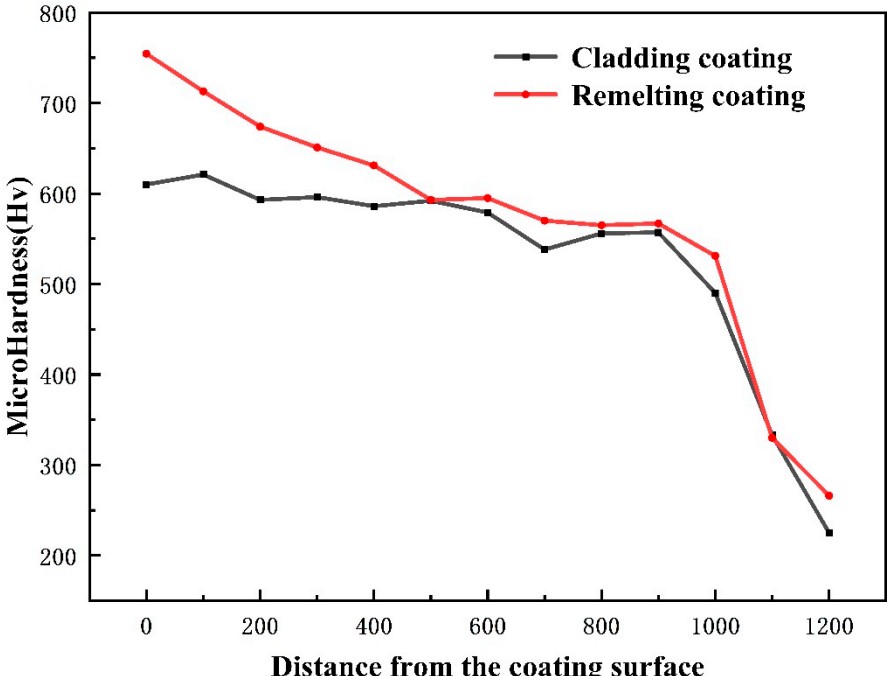

**Figure 9.** Hardness distribution across the coating cross-section.

Figure 10 shows the friction coefficients of the coating surface after 15 min of recipro-cating friction experiments. The laser-remelted coating exhibits a lower friction coefficient of 0.345, whereas the non-remelted coating has an average friction coefficient of 0.431. To provide a more intuitive representation of the wear resistance of the coating, measurement of the wear volume of the coating was conducted. A profilometer was utilized to measure the cross-sectional area (s) of the wear track along a direction perpendicular to the direction of the frictional wear, as illustrated in Figure 11. The measurement was performed three times, and the average value was calculated. The formula for calculating the wear volume (*V*) is as follows:

$$V = s \times h$$

where *h* is the wear scar length and *s* is the wear scar cross-sectional area.

Following the remelting treatment of the coating, there was a reduction in wear volume. The coating that underwent remelting exhibited a smaller wear volume of $2.55 \times 10^{-4}$ cm$^3$, whereas the unmelted coating displayed a larger wear volume of $7.45 \times 10^{-4}$ cm$^3$. It was observed that the wear resistance of the coating is directly proportional to its hardness. The laser remelting process can significantly enhance the wear resistance of the FeCoNiCrAl high-entropy alloy coating, possibly attributable to the enrichment of Al and Ni within the grain interiors after remelting, leading to the formation of hard phases.

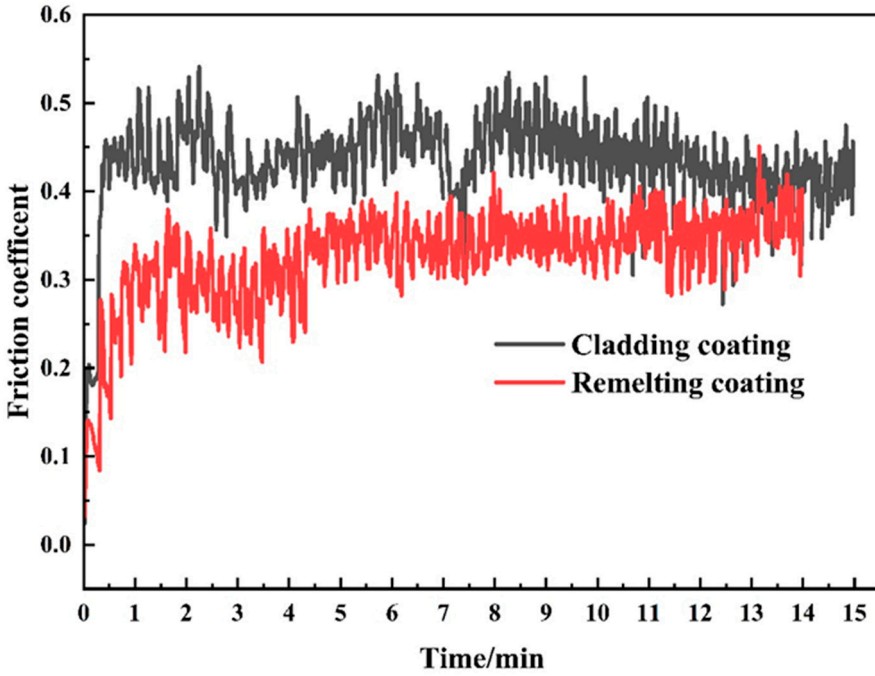

**Figure 10.** Friction coefficient.

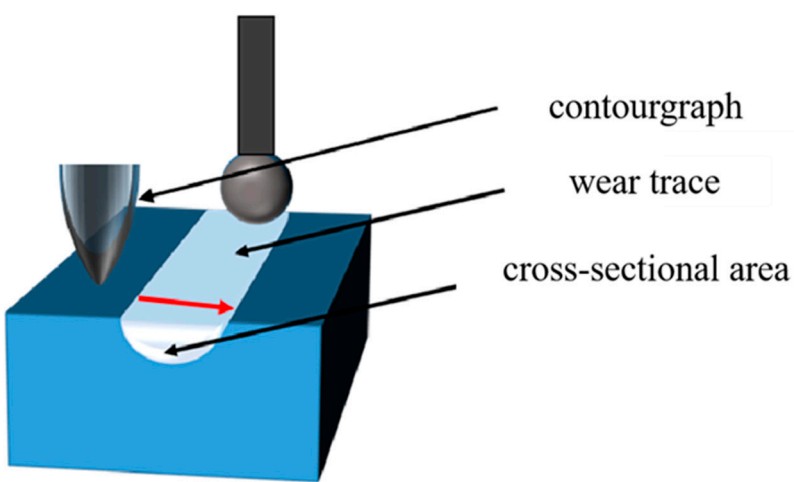

**Figure 11.** Schematic representation of wear experiment.

### 3.4. Corrosion Test

In order to investigate the impact of remelting on the corrosion resistance of the coating, electrochemical corrosion tests were conducted on both the substrate and the coating in a 3.5% wt NaCl solution. Figure 12 shows the potentiodynamic polarization curves for the coating and the Q235 substrate, and Tafel extrapolation was employed to determine the relevant parameters.

Figure 12 shows that the FeCoNiCrAl high-entropy alloy coating prepared by laser cladding exhibits the lowest corrosion current density, which is two orders of magnitude lower than the substrate. After remelting, the coating shows an increase in the corrosion potential, indicating a lower susceptibility to corrosion, but there is a slight increase in the corrosion current density. From the specific results of potentiodynamic polarization, as indicated in Table 3, it can be observed that the corrosion resistance of the coating is not significantly affected by laser remelting.

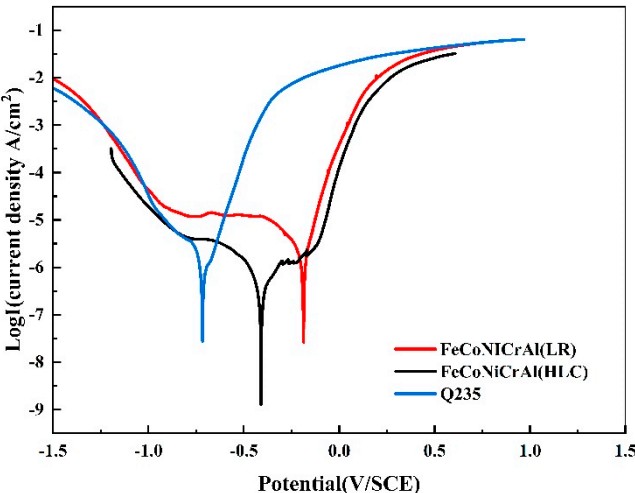

**Figure 12.** Potentiodynamic polarization curve.

**Table 3.** EDS results of FeCoNiCrAl high-entropy alloy coatings (At%).

|  | Fe | Co | Ni | Cr | Al |
|---|---|---|---|---|---|
| Cladding coating | 33.21 | 19.12 | 18.1 | 17.01 | 12.56 |
| Remelting coating | 26.02 | 18.03 | 19.63 | 16.01 | 20.31 |

To further validate the corrosion resistance of the remelted coating, electrochemical impedance testing was conducted. The alternating current impedance curve of the samples is depicted in the graph in Figure 13, which shows the Nyquist plot of the coating in a 3.5% wt NaCl solution. In the plot, $Z'$ represents the real part of impedance, and $Z''$ represents the imaginary part. The impedance modulus $Z$ is defined as:

$$|Z| = \sqrt{Z'^2 + Z''^2}$$

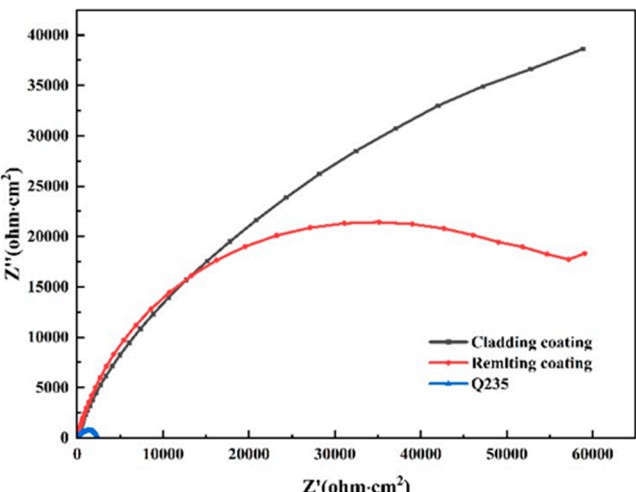

**Figure 13.** Nyquist plot.

The results, as shown in Figure 13, reveal that the HEA coating exhibits a larger capacitive arc in the high-frequency region, while the Q235 substrate's capacitive arc takes a semi-circular shape, albeit smaller in radius when compared to the coating. The remelting coating displays a significantly smaller capacitive arc radius than the cladding coating, approximately half of the cladding coating's radius. A larger capacitive arc radius indicates

a higher coating resistance, which can effectively protect the substrate from corrosion damage [24]. The corrosion resistance is ranked as follows: cladding coating > remelting coating > substrate.

Figure 14 represents a Bode plot showing the relationship between frequency and |Z|. In an electrochemical system, the system's resistance values change when alternating current is applied at various frequencies. Based on the above analysis, an equivalent circuit was constructed to describe the electrochemical reaction process. Figure 15 represents the equivalent circuit for the three samples in a 3.5% wt NaCl solution. In this circuit, Rs represents the electrolyte resistance and $R_{corr}$ stands for charge transfer resistance, depicting the transfer process between the clad layer and the substrate. To account for the non-uniformity of the electrode surface, Constant Phase Element (CPE) is often used in place of the Q element.

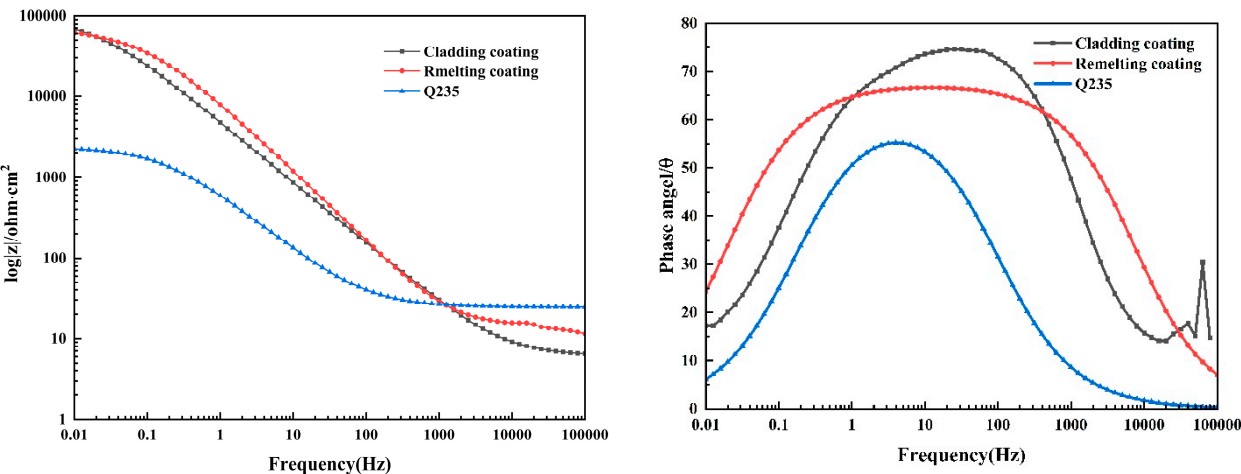

**Figure 14.** Bode plot.

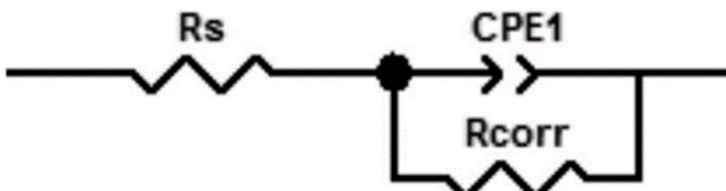

**Figure 15.** Equivalent circuit.

The resistance values can be calculated using the following equation:

$$Z = RS + \frac{1}{Y_0(j,\omega)^n}$$

where $\omega$ represents the angular frequency, $j$ is the imaginary unit, $Y_0$ is a frequency-independent constant, and n is the fitting exponent for the CPE ($n = 1$ represents ideal capacitive behavior and $n = 0$ represents pure resistance). The fitted data, as shown in Table 4, reveal that the charge transfer resistances for the Q235 substrate, the cladding coating, and the remelting coating are 2343 $\Omega \cdot cm^2$, 91,840 $\Omega \cdot cm^2$, and 59,540 $\Omega \cdot cm^2$, respectively. The charge transfer resistance is a parameter related to the corrosion rate, and the larger the $R_{corr}$ value, the lower the corrosion rate [25]. Therefore, this further confirms that the HEA coating exhibits excellent corrosion resistance compared to the substrate, and laser remelting did not significantly impact the coating's corrosion resistance.

**Table 4.** EIS fitting data.

|  | Q235 | Cladding Coating | Remelting Coating |
|---|---|---|---|
| $R_s$ | 24.37 | 6.105 | 15.17 |
| $Y_0$ | $4.01 \times 10^{-4}$ | $9.199 \times 10^{-5}$ | 2.65 |
| $n$ | 0.72 | 0.7498 | 0.8424 |
| $R_{corr}$ | 2343 | 91,840 | 59,540 |

*3.5. SVET Test*

SVET employs microelectrodes, signal converters, and amplifiers to reduce the disruption caused by microscans, thereby enhancing sensitivity and precision in measurements. The low destructiveness, high precision, and high sensitivity of SVET enable it to precisely gauge the electrochemical activity of samples in limited areas [26]. The SVET assay results, depicted in Figure 16, demonstrate the impact of immersing the laser-coated FeCoNiCrAl high-entropy alloy coating in a 3.5 wt% NaCl solution for 48 h, followed by the combination of 3D and 2D flat images. The 3D stereogram can be used to calculate the overall current density, and the SVET scan line can be used to determine the corrosion activity of the defect position, with the microanode and microcathode currents indicated by the peaks and troughs, respectively. The two-dimensional plane diagram illustrates the dispersion of the microanode and microcathode, indicating the diverse electrochemical reactions occurring on the coating's surface when chlorine ions are present. The degree of erosion of the coating by chlorine ions can be determined by analyzing SVET 3D and 2D floor plans. In situ observation of the FeCoNiCrAl high-entropy alloy coating in a 3.5% wt NaCl solution was conducted using SVET, revealing its local current density.

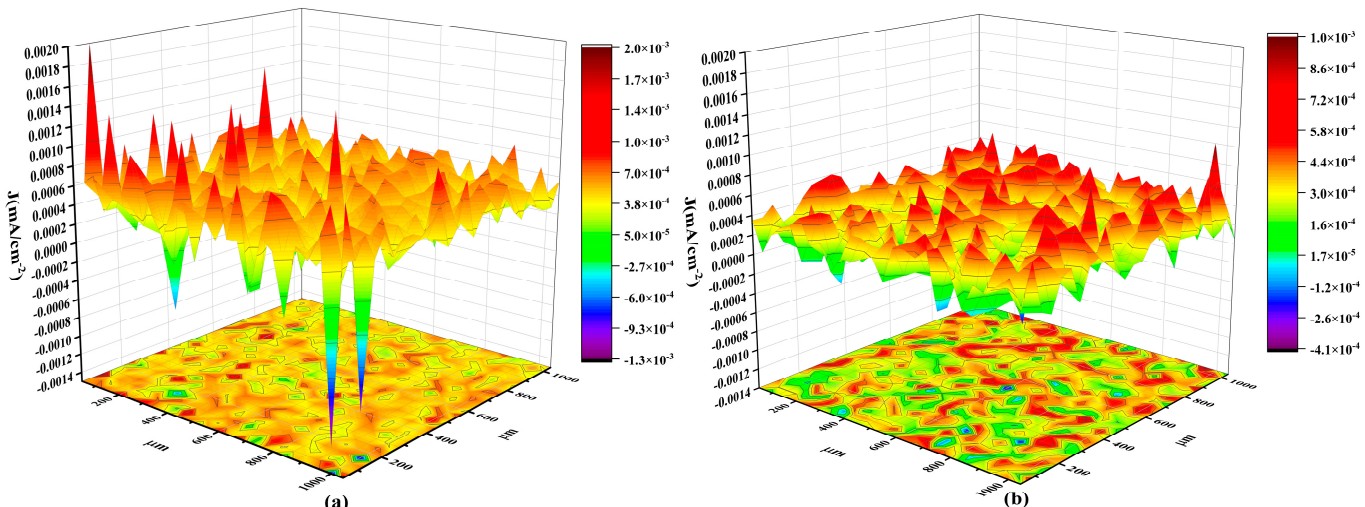

**Figure 16.** Scanning Vibrating Electrode Technique (SVET) testing: (**a**) cladding coating; (**b**) remelting coating.

The current density of the anode and cathode can be determined using the following equations to evaluate the corrosion behavior of the coating.

$$I_c = \int_0^{X \, max} \int_0^{Y \, max} [j_Z(x,y) < 0] dx dy$$

$$I_a = \int_0^{X \, max} \int_0^{Y \, max} [j_Z(x,y) > 0] dx dy$$

where $I_c$ is the cathode current density and $I_a$ is the anode current density.

The coating was immersed in a 3.5% wt NaCl solution for a continuous 48 h period. Upon analyzing the selected microregions, a significant elevation in the anodic peak was

observed in the edge region, accompanied by substantial fluctuations in current around the anodic peak. This is attributed to the presence of crack distribution in the edge region. In contrast, another coating subjected to laser remelting exhibited a uniform distribution of anodic current. SVET utilizes a microprobe to examine the material's surface and determine the dispersion of surface currents. The size of the surface current is related to the corrosion reaction and state of the FeCoNiCrAl high-entropy alloys coatings. The negative current is related to the cathode reaction and the positive current is related to the anode reaction. The results of applying the conversion formula are displayed in Table 5.

**Table 5.** Anodic current density.

|  | $I_a/(\mu A \cdot cm^{-2})$ |
| --- | --- |
| Cladding coating | 0.52 |
| Remelting coating | 0.63 |

## 4. Discussion

### 4.1. Phase Formation and Evolution

The thermodynamic parameters $\Delta H_{mix}$ (enthalpy of mixing), $\delta$ (atomic radius difference), $\Delta S_{mix}$ (mixed entropy), and $\Omega$ can be used to forecast the formation and structural stability of HEAs [27]. The formation of the solid solute phase [28] was contingent upon the fulfillment of $-15$ kJ mol$^{-1} \leq \Delta H_{mix} \leq 5$ kJ mol$^{-1}$ and $\delta \leq 6.5\%$) as well as the $\delta \leq 6.6\%$ criteria $\Omega \geq 1.1$. Furthermore, Guo et al. [29] determined that a greater abundance of valence electrons ($VEC$) ($\geq 8$) was advantageous for the formation of FCC solid solutions, whereas a smaller $VEC$ ($<6.87$) was advantageous for the formation of BCC solid solutions. This is the equation for the calculation:

$$\Delta H_{mix} = 4 \sum_{i=1, i \neq j} \Delta H_{ij} c_i c_j$$

$$\Delta S_{mix} = -R \sum_{i=1}^{n} c_i \ln c_i$$

$$\Omega = \frac{T_m \Delta S_{mix}}{|\Delta H_{mix}|}$$

$$\delta_r = 100 \sqrt{\sum_{i=1}^{n} c_i \left(1 - \frac{r_i}{\bar{r}}\right)^2}$$

$$\bar{r} = \sum_{i=1}^{n} c_i r_i$$

$$T_m = \sum_{i=1}^{n} C_i (T_m)_i$$

$$VEC = \sum_{i=1}^{n} c_i (VEC)_i$$

where $c_i$ and $c_j$ are the atomic fraction of elements $i$ and $j$ in the alloy, $R$ is the gas constant, $(T_m)_i$ is the melting point of element $i$, $r_i$ represents the radius of the element $i$, and $\bar{r}$ is the average atomic radius.

The parameters of FeCoNiCrAl high-entropy alloys coatings were calculated as shown in Table 6. After laser remelting treatment, the VEC value of the FeCoNiCrAl high-entropy alloy coating decreased from 7.81 to 7.12, and the $\delta$ value increased from 3.92 to 4.81. A small VEC was more conducive to the formation of a BCC structure. The BCC structure could adapt to large d values with a lower strain energy penalty. The solidification rate in laser remelting (typically 105 K/s [30]) is much slower than that in laser cladding (~106 K/s [31]). As a result, the elements in the coatings migrated significantly, with an

increase in the percentage of Al elements in the coatings and a decrease in the percentage of Fe elements, leading to the transformation of the coatings from a mixture of FCC, A2, and B2 to a mixture of A2 and B2.

**Table 6.** Thermodynamic parameters.

| Alloy | $\Delta H_{mix}$ (kJ·mol$^{-1}$) | $\Delta Smix$ (J·K$^{-1}$ mol$^{-1}$) | $T_m$ (K) | $\Omega$ | $\delta$ (%) | *VEC* |
|---|---|---|---|---|---|---|
| Cladding coating | −7.61 | 12.45 | 1745.26 | 2.86 | 3.92 | 7.81 |
| Remelting coating | −9.92 | 12.82 | 1715.32 | 2.216 | 4.81 | 7.12 |

### 4.2. Effect of Laser Remelting on the Wear Resistance of Coatings

The process of preparing FeCoNiCrAl high-entropy alloy coatings using laser cladding on the surface of Q235 base material is prone to defects such as porosity cracks, which tend to promote the emergence and expansion of cracks, thus seriously affecting the wear-resistant properties of the coatings. After remelting treatment, these defects in FeCoNiCrAl coatings basically disappeared, and the wear resistance of the coatings was improved.

The surface analysis indicated a higher proportion of Fe elements and a lower proportion of Al elements on the coating surface. After undergoing laser remelting treatment, there was an increase in the proportion of Al elements on the coating surface. Research has demonstrated that the elevated proportion of Al elements contributes to an improvement in the coating's hardness [32], consequently enhancing its wear resistance.

### 4.3. Effect of Laser Remelting on the Corrosion Resistance of Coatings

Laser remelting proved effective in eliminating surface defects, such as cracks and pores, thereby reducing the inherent corrosion susceptibility of the coating. Nevertheless, despite the positive impact on surface integrity, laser remelting did not yield a substantial enhancement in the corrosion resistance of the coatings when subjected to NaCl solution, as evidenced by both macroscopic electrochemical tests involving kinetic potential polarization and electrochemical impedance.

Analysis of the Scanning Vibrating Electrode Technique (SVET) plots revealed a pronounced anodic current peak in the cracked region of the coating (Figure 17a), contrasting with the low anodic current observed in the non-cracked region (Figure 17b). The uniform distribution of anodic current on the coating's surface following remelting played a pivotal role in mitigating the higher anodic current peaks associated with cracks. The average current density in cracked and non-cracked areas, presented in Table 7, indicated a significantly higher anodic current density in the cracked region.

**Table 7.** Anodic current density.

| | $I_a$/(μA·cm$^{-2}$) |
|---|---|
| Cladding coating (crack area) | 1.257 |
| Cladding coating | 0.424 |
| Remelting coating | 0.63 |

The corrosion morphology of the cracked area of the fusion cladding, as depicted in the Figure 18, illustrated crack expansion along the vertical plane. The confined space within the crack hindered oxygen diffusion, resulting in oxygen deficiency inside the crack. This led to the formation of a concentration battery, with the higher oxygen concentration acting as the cathode and the lower oxygen concentration as the anode, facilitating anodic dissolution within the crack.

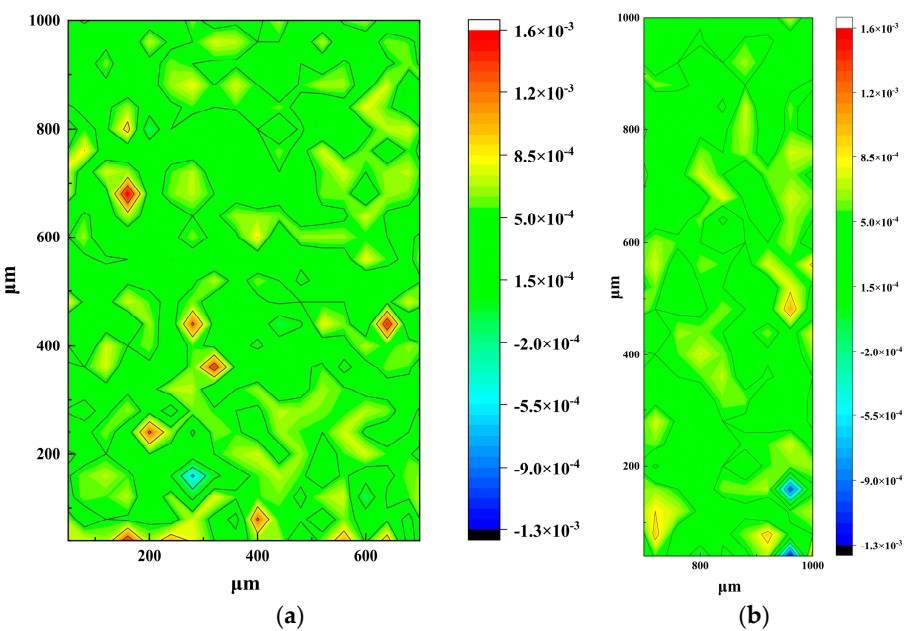

**Figure 17.** Scanning Vibrating Electrode Technique (SVET) Testing: (**a**) cracked region; (**b**) non-cracked region.

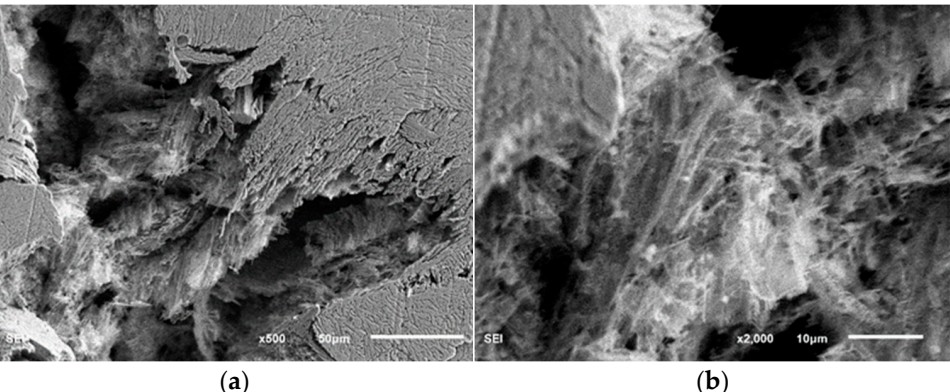

**Figure 18.** Corrosion morphology of the crack region: (**a**) outside of the crack; (**b**) inside of the crack.

Conversely, Figure 19 illustrates that the coating surface post laser remelting exhibited a lack of crack defects, an absence of evident corrosion pits, and only a few isolated pitting holes. Thus, while laser remelting significantly improved the corrosion resistance of the cracked areas, its impact on crack-free regions was not significant.

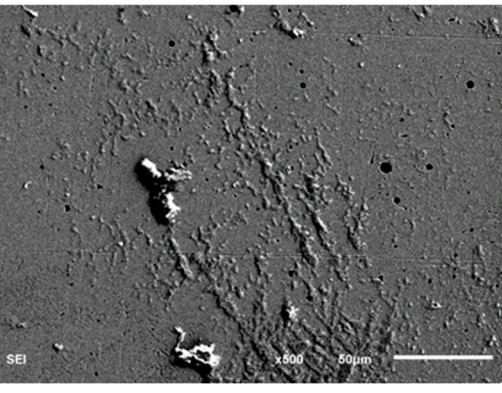

**Figure 19.** Corrosion morphology of remelting coating.

## 5. Conclusions

In this study, laser cladding was used to prepare FeCoNiCrAl high-entropy alloy coatings, and laser remelting was performed. The study examined the influence of laser remelting on the phase composition, microstructure, and coating properties, including wear resistance and corrosion resistance.

Following laser remelting treatment, the atomic proportion of Fe elements on the coating surface decreased from 33.21% to 26.03%, while the atomic proportion of Al elements increased from 12.56% to 20.31%. The phase composition of the coating underwent a marked transformation, shifting from a structure composed of FCC, A2, and B2 phases to a singular BCC structure characterized by the presence of A2 and B2 phases. Concurrently, the grain morphology on the coating surface transitioned from elongated plate-like grains to equiaxed grains.

After remelting, the hardness of the AlCoCrFeNi coating changed from 613 to 754 $HV_{0.2}$, an increase of 23%. The coefficient of friction on the coating surface decreased from 0.431 to 0.345, resulting in a 65% reduction in wear volume.

The FeCoNiCrAl coating exhibits excellent corrosion resistance, and laser remelting repairs defects such as cracks, further enhancing the corrosion resistance of the coating. However, it has no significant impact on the corrosion resistance of non-cracked regions.

Further investigation is necessary to determine the structure and characteristics of coating when using a single laser power to remelt the surface under various remelt powers, as certain restrictions exist.

**Author Contributions:** The experiment was designed by T.L. and W.Z. Methodology, X.J. The result analysis was performed by T.L. J.H., W.Z. and C.Z. were responsible for writing the paper. All authors have read and agreed to the published version of the manuscript.

**Funding:** This research received no external funding

**Data Availability Statement:** Data are contained within the article.

**Conflicts of Interest:** The authors declare no conflict of interest.

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
