# Peer review of "Study on the Microstructure and Properties of FeCoNiCrAl High-Entropy Alloy Coating Prepared by Laser Cladding-Remelting"

_coatings, doi:10.3390/coatings14010049_

Round 1

Reviewer 1 Report

Comments and Suggestions for Authors

Author Response

Dear Reviewer:

     Thank you for taking the time to carefully review my paper. I have made all the revisions as per your suggestions. Please refer to the attached section for details

Reviewer 2 Report

Comments and Suggestions for Authors

The present paper is an investigation into the Microstructure and Properties of Laser-Clad and Remelted FeCoNiCrAl High-Entropy Alloy Coating.The investigation is centered on scrutinizing the influence of laser remelting on the structure and properties of high-entropy alloy coatings. After laser remelting, several notable improvements are observed:

  • Porosity cracks disappear from the coating's surface.
  • The grain structure becomes finer.
  • Columnar crystals transform into equiaxed crystals.

Notably, Aluminum (Al) and Nickel (Ni) are primarily distributed within the grains, while Iron (Fe) and Chromium (Cr) accumulate at the grain boundaries. These changes in microstructure and elemental distribution are key findings that shed light on the impact of laser remelting on the properties of high-entropy alloy coatings. This study's topic is up-to-date but needs improvement before its publication.  

First, the authors need to improve the abstract. In its present form, it is too lengthy. 

In the introduction section, the authors need to add more relevant literature. 

The authors should indeed provide clarity on how they selected the specific alloy and its precise composition for their study. Such information is crucial for readers to understand the basis for their research and the rationale behind their choice of materials.

The authors need to add a discussion section. A discussion section allows the authors to contextualize their results, identify trends or differences, and contribute to the broader scientific understanding of the topic. 

Author Response

Dear Reviewer:

Thank you for taking the time to carefully review my paper. I have made all the revisions as per your suggestions. Please refer to the attached section for details.

Reviewer 3 Report

Comments and Suggestions for Authors

The manuscript has the following queries to be answered.
1. Line no. 148-150 needs proper evidence and references.
2. What is the crystal structure of B2 phase.
3. Figure 4, 5(a, b) need to be replaced with high magnification good qualities.
4. Experimental proof for elemental distribution mentioned in Line no. 168-172 is required.
5. Unit of microhardness needs to be properly mentioned.
6. Most of the results are mentioned without any proper discussion.
7. Effectiveness of laser surface remelting should be mentioed in the introduction with citing the paper "Enabling plastic co-deformation of disparate phases in a laser rapid solidified Sr-modified Al–Si eutectic through partial-dislocation-mediated-plasticity in Si".

8. Major issue with the manuscript is that there is no discussion section involved in the manuscript.

Comments on the Quality of English Language

The manuscript is full of grammatical errors.

Author Response

(The authors gave the same response as above.)

Round 2

Reviewer 1 Report

Comments and Suggestions for Authors

I have reviewed the manuscript Study on the Microstructure and properties of FeCoNiCrAl high entropy alloy coating prepared by laser cladding-remelting. The authors have answered all the questions and have changed the manuscript accordingly. Therefore, I recommend the manuscript for publication in the MDPI-Coatings.

Reviewer 2 Report

Comments and Suggestions for Authors

All queries raised have been satisfactorily addressed by the authors, and the manuscript has been revised as needed. Consequently, I endorse its publication in the MDPI-Coatings journal.

Reviewer 3 Report

Comments and Suggestions for Authors

The referencing should be followed as per the journal guidelines. Overall, the revised version is in good shape for publication.

Comments on the Quality of English Language

No comments